# Stabilisation of Pathologic Proximal Femoral Fracture near the Growth Plate with Use of a Locking Plate and Transphyseal Screws

**DOI:** 10.3390/children9121932

**Published:** 2022-12-09

**Authors:** Roman Michalik, Frank Hildebrand, Heide Delbrück

**Affiliations:** Department of Orthopaedic, Trauma and Reconstructive Surgery, University Hospital RWTH Aachen, D-52074 Aachen, Germany

**Keywords:** bone cyst, paediatric orthopaedics, osteosyntheses in children, growth plate, transphyseal screws

## Abstract

Aneurysmal bone cyst (ABC) is a benign osseus lesion with a high pathologic fracture risk. The described treatment options are varied and inconsistent. For successful treatment results, it is essential to prevent recurrence and sufficiently stabilise the weakened bone. Lesions close to the growth plates, especially in the femoral neck region, are challenging to stabilise in children. In this study, 27 clinics, including 11 sarcoma centres, 15 paediatric orthopaedic clinics, and one sarcoma/paediatric orthopaedic centre, were surveyed and asked about their treatment approaches for an exemplary case of ABC in the femoral neck causing a pathological fracture in a 20-month-old infant, with a response rate of 81%. The heterogeneity of treatment options described in the literature is consistent with the survey results. The most favoured approach was curettage, defect filling of any kind, and surgical stabilisation. However, the lesion stabilisation option introduced in this paper, which involves the use of transphyseal screws, was not mentioned in the survey and has not been reported in the literature. Contrary to the existing concepts, our technique offers high stability without significant growth restriction. Transphyseal screws are also suitable for the treatment of femoral neck fractures of other aetiologies in children.

## 1. Introduction

Aneurysmal bone cyst (ABC) is a benign tumorous lesion. Approximately 70% of lesions are found in patients between the ages of 5 and 20 years, with little predilection to the female sex [1,2]. The lesion can occur in almost any bone, although the long bones are preferentially affected [3,4]. Mankin et al. analysed 150 cases of ABC over 20 years, with the lesions occurring predominantly in the tibia, femur, fibula, pelvis, humerus, clavicle, foot, and lumbar spine, in descending order [5]. The lesion occasionally becomes symptomatic with local swelling or pain but frequently presents with a pathological fracture due to the thinned cortical bone. Various treatment methods are available, as the high risk of recurrence, which has been described in up to 60% of cases in 24–50 months [6], has constantly led to the development of new approaches.

For the therapy of the cyst itself, bloc resection of the lesion, radiation, sclerotherapy (e.g., with polidocanol), and curettage are used alone or in combination. In a prospective randomised study, Varshney et al. investigated the effectiveness of repeated polidocanol sclerotherapy in comparison with that of curettage in combination with spongiosaplasty and additional filling with synthetic hydroxyapatite. The authors concluded that, although the therapeutic strategies had equal healing rates, recurrent sclerotherapy had the advantages of outpatient feasibility, faster pain reduction, and better functional outcomes [7]. However, other studies have advocated the open procedure using curettage and, if necessary, cancellous bone or cement filling [5,8,9].

Cure rates increase with the number of procedures. In addition to therapy for the cystic lesion itself, stabilisation of the bone is often required. A challenge here is the localisation in the femoral neck and proximal femur region. In epiphyseal localisations of the heavily loaded femoral neck and proximal femur, osteosynthesis using the usual procedures of elastic stable intramedullary nailing (ESIN) and Kirschner wires during the growing age is limited. In addition, immobilisation in a hip spica cast (HSC) would result in complete immobilisation of the child for weeks.

In this work, we present a surgical one-stage treatment of an ABC with a pathologic fracture at the proximal femur and femoral neck in a 20-month-old patient. We present the case in a nationwide survey among certified German sarcoma centres and children orthopaedic departments to determine their preferred diagnostics and treatment. In our surgical approach, we performed osteosynthesis using a locking plate with so-called transphyseal screws, which could cross the growth plate at the femoral neck owing to their shape (metaphyseal thread with smooth threadless epiphyseal shape).

## 2. Materials and Methods

### 2.1. Case Report

A 20-month-old boy presented with a limping gait pattern and sparing of the left leg. Radiological examination revealed osteolysis of the femoral neck and proximal femur, adjacent to the growth plate, with a typical multiseptated aspect, marked cortical weakening, or interruption in the axial image. Magnetic resonance imaging (MRI) clearly revealed the mirror formation of the cyst contents (Figure 1).

In the otherwise healthy patient, after a biopsy and an intraoperative frozen section investigation, a one-step surgical treatment was performed by subsequent curettage of the cyst (Figure 2A–C), irrigation with polidocanol, filling with allogen cancellous bone, and subsequent osteosynthesis using a locking proximal femoral plate (OrthoPediatrics PediLoc) and transphyseal locking screws (OrthoPediatrics Corp., Warsaw, IN, USA; Figure 2D–F).

The follow-up treatment could be designed without a cast with pain-dependent mobilization.

At the first outpatient visit, 6 weeks after the operation, the boy presented without pain, with an unremarkable gait pattern, and with unrestricted hip mobility. He had already returned to kindergarten. The final immuno-histopathological results confirmed the suspected diagnosis of ABC. A translocation of the *GNP6* gene was found, which is known to be linked to primary ABC [10]. The radiological examination revealed good cyst filling and screw position (Figure 3A,B). Six months after the operation, the screw tips, which were originally located in the epiphysis, had almost migrated towards the metaphysis, but a recurrence was suspected (Figure 3C,D).

However, the patient still had no pain and an unremarkable clinical examination finding. Therefore, a short-term clinical and radiological control was agreed, which showed no significant cyst progression even after 3 months; thus, further follow-up controls were scheduled (Figure 3E,F). No evidence of growth disorders or femoral head necrosis was found. There is also no need to assume any risk to stability if there is no pain and the screws are in place and thus no urgent need for action.

### 2.2. Conducted Survey

The anonymised preoperative images (Figure 1A–D) were sent with the abovementioned medical history to the medical directors of all 11 German sarcoma centres certified by the German Cancer Society (www.oncomap.de) at that time (accessed on 27 August 2021). They were also sent to the heads of all 16 children orthopaedic departments listed in “KlinikKompass” (www.klinikkompass.com, accessed on 27 August 2021) as the best clinics for paediatric orthopaedics in Germany. This is a platform founded in November 2018 by a journalist to help find a specialist clinic. This list was based on high case numbers for the searched field, patient safety, and, according to the surveys by the health insurance companies, high patient satisfaction. The text of inquiry was the following: “Attached are pictures of a 20-month-old boy who was already walking safely but now has a clear limp and left leg rest. We are contacting you because you are considered a certified sarcoma centre/leading clinic for paediatric orthopaedics in Germany, and in this case, there are certainly several possible approaches. How would you proceed or treat the existing bone cyst?” The entity of the lesion was not known at this time. The corresponding survey results were analysed with the consent of the participating clinics. The data protection regulations were complied with.

## 3. Results

### 3.1. Survey Results

A nationwide survey was conducted among 11 sarcoma centres, 15 paediatric orthopaedic clinics, and one sarcoma/paediatric orthopaedic centre in Germany. A total of 22 clinics gave their assessments (response rate, 81%).

#### 3.1.1. Entity Assessment

To the question about entity estimation, 18 clinics responded. On the basis of the provided native radiological images and MRI scan, a suspicion of ABC (36.4%) was predominantly expressed, and a differential diagnosis of juvenile bone cyst (JBC; 22.7%) or telangiectatic osteosarcoma (OS; 4.5%) was also mentioned. The sole suspected diagnosis of a juvenile bone cyst (JBC) was expressed by 18% of the hospitals (Figure 4A).

#### 3.1.2. Assessment of the Need for Biopsy

An additional biopsy confirmation of the diagnosis before initiating therapy was requested by most clinics (59.1%), 9% recommended an MRI follow-up as an alternative, 22.7% denied a biopsy, and 18.2% did not provide any information (Figure 4B).

#### 3.1.3. Entity-Specific Local Therapy

The answers to the question about entity-specific therapy were more heterogeneous (Figure 5A). Most hospitals recommended a primary surgical procedure (72.7%). Only 22.7% of the clinics preferred adjuvant injection in the cyst alone. Some respondents (4.5%) stated that the healing of the cyst is stimulated by the fracture that has occurred and that a wait-and-see behavior can be adopted.

Curettage was predominantly mentioned as a therapy for the cyst (68.2%). In 4.5% of cases, the therapy was extended with an adjuvant injection. In 4.5% of cases, curettage was mentioned as a therapy in the event of persistent findings after immobilisation using an HSC. Further treatment options were sole instillation (22.7%) of an adjuvant such as polidocanol (4.5%) or phenol (4.5%) for a suspected ABC or methylprednisolone acetate (9%) for JBC. In the case of biopsy-proven osteosarcoma, a Borggreve-Van Nes-Winkelmann rotationplasty was recommended as a treatment option (4.5%).

#### 3.1.4. Defect Filling

Regarding defect filling (Figure 5B), 77% of the 22 participating clinics gave their assessment, with the remaining having either previously recommended conservative therapy (13.6%) or not provided any information (9%).

Filling with allogen spongiosa alone (13.6%) or in combination with bone replacement material (13.6%) or autologous bone (4.5%) was mentioned with a similar frequency as the recommendation to use bone replacement material alone (27.3%) or fibula interposition with/without bone replacement material (9%). Among the respondents, 9% recommended a finding-dependent procedure with allogen spongiosa or bone substitute material. A multiple-stage procedure using a bone cement (Palacos) seal and, after an observation interval, a secondary filling with spongiosa was also recommended by 9% of the participating clinics.

#### 3.1.5. Stabilisation/Osteosynthesis

Of the responding clinics, 40.9% clearly recommended stabilisation (Figure 5C); 13.6% preferred osteosynthesis-free treatment, with an HSC being the most recommended; 9% recommended the use of an HSC as an alternative to surgical treatment with a plate osteosynthesis. Osteosynthesis with a plate was mainly recommended (27%), whereby the classic locking screws should be inserted over the growth plate (GP). Retrograde ESIN (9%) was suggested as an alternative, with one clinic recommending ESIN and a plate. A GP-bridging osteosynthesis using Kirschner wires and additional application of an HSC was recommended as a treatment procedure by 4.5% of the hospitals.

## 4. Discussion

From our point of view, the presented case of ABC that caused a pathological fracture in the femoral neck and proximal femur region in a 20-month-old boy is unique in several aspects: First, ABCs are usually found in children at ages between 5 and 20 years [1,2]; thus, the presented case of ABC in a 20-month-old boy is rare. Other differential diagnoses must also be considered initially. Furthermore, because of the large extent of the lesion, including the hole femoral neck and proximal femur, and its localisation up to the growth plate of the femoral neck, treatment was challenging, as in addition to treating the lesion, the pathological fracture, which was in the immediate vicinity of the growth plate, must also be stabilised.

Owing to the mentioned peculiarities of this case and as treatment strategies for bone cysts, especially ABCs, are inconsistent, we asked paediatric orthopaedic specialist clinics and sarcoma centres for recommendations on how to proceed as part of an open inquiry. The high response rate of our survey of 81% shows the interest and the need for discussion of this topic and our presented case. Considering the survey results and discussing the treatment options with the patient’s parents, we chose a one-stage treatment approach consisting of a biopsy and intraoperative frozen section investigation (for exclusion of malignancy and confirmation of the suspected diagnosis of ABC), extension of the approach, subtle curettage, polidocanol injection, defect filling with allogenous spongiosa, and osteosynthesis with a locking plate and the described special transphyseal screws.

The diagnosis of ABC can be made based on the combination of conventional radiography and MRI, with almost 83% sensitivity and 70% specificity [11]. The imaging diagnostic findings of our case, which showed multiple septation and mirror formation on MRI, were highly indicative of ABC. However, in our survey, only 36.4% favoured this diagnosis, whereas 27.2% considered ABC as a differential diagnosis on the basis of the presented radiography and MRI results. As mentioned earlier, only one third of the respondents suspected ABC as a diagnosis possibly because of the untypical young age of the patient. Furthermore, we suspected that highly specialised clinics want to avoid misdiagnosis in cases where suspected lesions are initially diagnosed as benign but are actually malignant. This has certainly affected a high number of respondents who aimed for a biopsy prior to further treatment (59.1%). Only 22.7% of the respondents did not consider a biopsy of the lesion. To reduce surgical burden, we chose an intraoperative frozen section analysis, which none of the interviewees mentioned as a possibility. Their reason could be the limited value of the use of frozen sections in the detailed entity classification of bone lesions. However, on the basis of contradicting study results, we believe that a first assessment of the dignity can be expected [12,13].

Regarding the treatment of the cyst itself, we decided to perform a curettage because of bone instability and the large extent of the lesion. Thus, we could immediately fill the defect with allogenous spongiosa to provide a more stable bone stock. Instillation of polidocanol as a sole therapy seemed inappropriate, although sequential percutaneous instillations of polidocanol are seemingly equally effective as intralesional curettage in the therapy of primary ABCs [7,14]. The repeated necessary procedure is considered an apparent disadvantage of polidocanol instillation. The apparently more aggressive open surgical procedure that we performed was also appropriate owing to the planned additional plate osteosynthesis. We also dabbed the lesion with polidocanol before filling. To our knowledge, the outcome of the combination of curettage and polidocanol instillation has not been evaluated. Furthermore, possible growth arrest or delay has to be kept in mind when using sclerotherapy near the growth plates because injection site necrosis was previously reported [7].

Regarding defect-filling bone substitutes, in our survey, cement or spongiosa was proposed for use in approximately equal parts to fill up the cyst. This result is representative of the reports in the literature and corresponds to the recommendations of the German S1 guideline “Bone cysts” [15]. Moreover, the combined use of allogenic bone grafting and artificial bone graft substitutes was reported by Tomaszewski et al. [16]. We decided to fill the defect in our case with allogenic spongiosa alone because the advantages with regard to consolidation and healing of the bone stock of one of the mentioned substitutes or compositions have not been completely evaluated [17].

Most of the 22 respondent hospitals recommended surgical stabilisation with a plate or intramedullary nails. Three clinics preferred immobilisation with a cast alone. Stabilisation techniques of the weakened bone in comparable situations have been rarely discussed in the literature. In their study, Vergel De Dios et al. examined 238 ABCs, with the femur (40 patients) being the most frequently affected among the long bones; the proximal femur was involved in 10 cases [18]. The authors identified “numerous forms of therapy” independent of localisation, including curettage and radiation but no stabilisation techniques. In the recently published retrospective multicentre EPOS study, in which 79 patients with ABC on the proximal femur were examined, no detailed information on the selection of the osteosynthesis technique, especially in the immediate vicinity of the growth plate, was provided [19]. The transphyseal screws we used were also not mentioned. Tomaszewski et al. reported their results from 30 children with bone cysts, tumour-like lesions of the proximal femur region, and pathological fractures [16]. They also used a locking or angular plate for stabilisation but did not mention the use of transphyseal screws.

As an open procedure was chosen by 43% of our respondents and 44% of the participants of the EPOS multicentre study [19], in our opinion, there is nothing to be said against a plate osteosynthesis, which does not significantly exceed the extent of the intervention. From our point of view, immobilisation using an HSC offers, at best, an alternative if the procedure is not open. However, above a certain lesion size, the extent at which the usual wearing time of 6 weeks is sufficient is questionable, as bone consolidation cannot be assumed within this time in the case of the present entity and lesion size. Retrograde ESIN as an alternative can hardly bridge the growth plate. Kirschner wires alone do not provide sufficient stability in an epi-meta-diaphyseal position.

Locking plate osteosynthesis, or a blade plate, is used as a more stable alternative. This is used for corrective osteotomies of the intertrochanter region in elective orthopaedics in children and can also be implanted in the event of a fracture. However, if the fully threaded locking screws used for this purpose are passed over the growth plate, growth disorders are to be expected [20,21].

The transphyseal locking screws used are characterised by a thread-free zone at the distal end of the screw, which enables length growth in this area and offers the ideal implant for the case described, such as femoral neck fractures of all origins in children with an open growth plate. None of the clinics suggested the implant in the initial survey, and the implant was not described in the above-mentioned studies. According to the current state of knowledge, growth disorders, femoral head necrosis, or biomechanical problems are not to be expected from the implant itself, as the plate design corresponds to that of the plates used for corrective osteotomies and the end of the screws corresponds to that of Kirschner wires. The benefits of immediate pain-adapted loading are significant. The disadvantage that must be mentioned is that the screws and plates are made of steel, which makes follow-up examination using MRI unsuitable. However, because cyst recurrences that require revision are sufficiently visible on conventional radiography, this aspect does not seem to be of primary importance to us.

This study has several limitations. Although in our case, we can report a good treatment result in the short term without radiologically detectable growth disturbances of the femoral epiphysis and femoral neck, longer follow-up studies and a larger number of cases in different age groups are necessary to provide definitive conclusions on the unrestricted use of transphyseal screws. Longer follow-up is needed to determine whether a progredient cyst recurrence will occur in turn requires follow-up operations that would lead to growth problems caused by the intervention but also by the disease itself.

## 5. Conclusions

As part of the treatment of bone cysts in the femoral neck, near the open growth plate, and femoral neck fractures of other origins, stable osteosynthesis with transphyseal screws and a locking plate should be considered. This method of osteosynthesis was obviously not widely known in the context of our survey of specialised hospitals. In addition, we advocate the feasibility of a one-stage procedure for cystic bone lesions in children, accompanied with an intraoperative frozen section investigation when in doubt of the suspected histology, which, however, requires the operation to be performed in clinics with the appropriate expertise and equipment. 

## Figures and Tables

**Figure 1 children-09-01932-f001:**
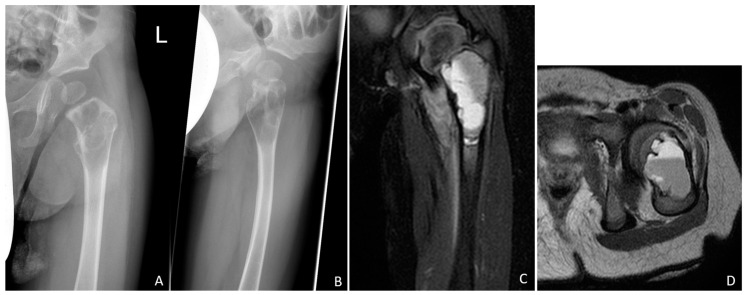
Conventional radiographs of the left hip joint showing the septated cystic lesion of the proximal femur extending immediately to the growth plate in two planes (**A**,**B**). Perforation of the thinned cortical bone is indicated in the lateral image (**B**). The magnetic resonance image clearly shows the multichambered fluid-filled cyst, which is confined to the proximal femur (**C**,**D**). In the axial section, the fluid levels are clearly visible (**D**).

**Figure 2 children-09-01932-f002:**
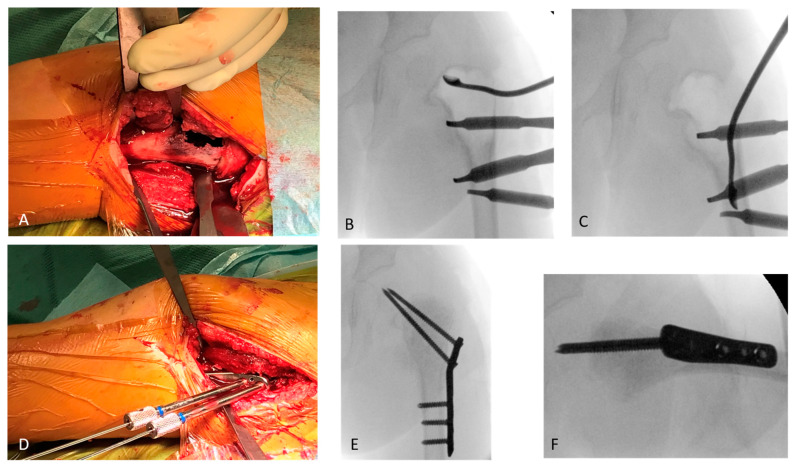
The femur is reached via a lateral approach. The cyst has already perforated the ventral cortex (**A**). The cyst is excochleated with a flexible sharp curette, which is documented with the image intensifier (**B**,**C**). After exclusion of malignancy in the frozen section and subsequent excochleation, rinsing with polidocanol and filling of the cyst with allogen spongiosa an osteosynthetic stabilisation is performed (**D**). After completion of the osteosynthesis, the X-ray control is carried out in two planes (**E**,**F**), showing the correct central position of the transphyseal screws in the epiphysis.

**Figure 3 children-09-01932-f003:**
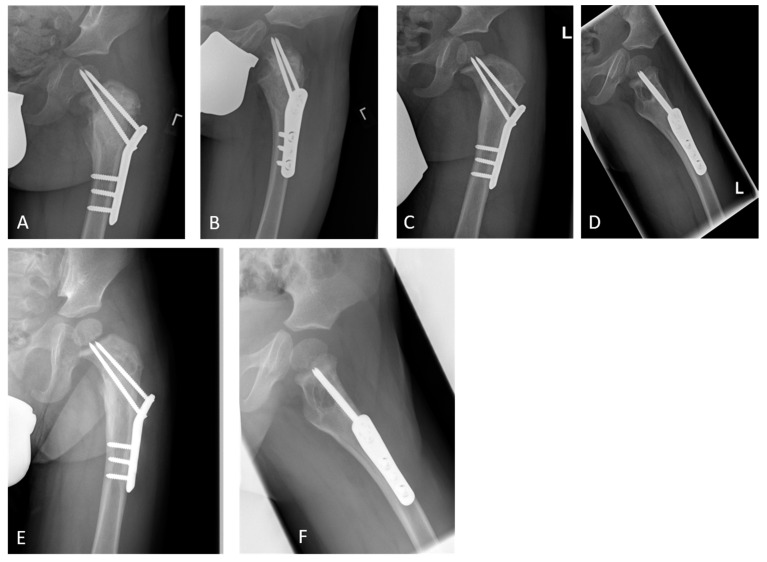
Radiological follow-up examination in the short and medium terms. After 6 weeks (**A**,**B**), the desired result was achieved, with a regular material position and filled cyst. The growth of the proximal femur over the physeal screws can been seen clearly in the displayed radiographs (**A**–**F**). However (**C**,**D**), radiological evidence suggests a recurrence, with the findings remaining stable for another 4 months (**E**,**F**).

**Figure 4 children-09-01932-f004:**
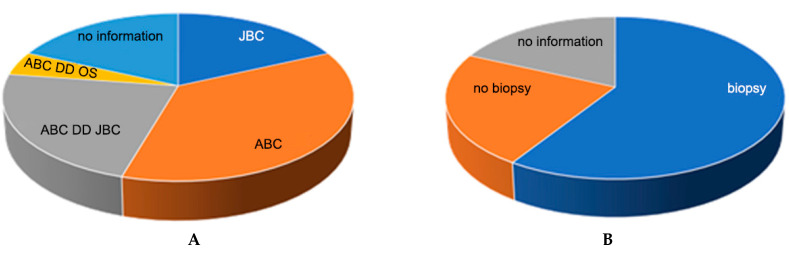
(**A**) Entity assessment: Most participating clinics (n = 22) classified the lesion as an aneurysmal bone cyst (ABC; 36.4%) or considered it in the differential diagnosis (ABC, DD, or juvenile bone cyst [JBC]: 22.7%; ABC, DD, or osteosarcoma (4.5%). Thus, JBC (n = 4) was the next most frequently mentioned differential diagnosis or sole diagnosis. Overall, four responding hospitals did not provide any information on this question. (**B**) Regarding the need for biopsy, of the responding hospitals, 59.1% wanted a biopsy, 22.7% did not consider it necessary, and 18.2% did not provide any information on this.

**Figure 5 children-09-01932-f005:**
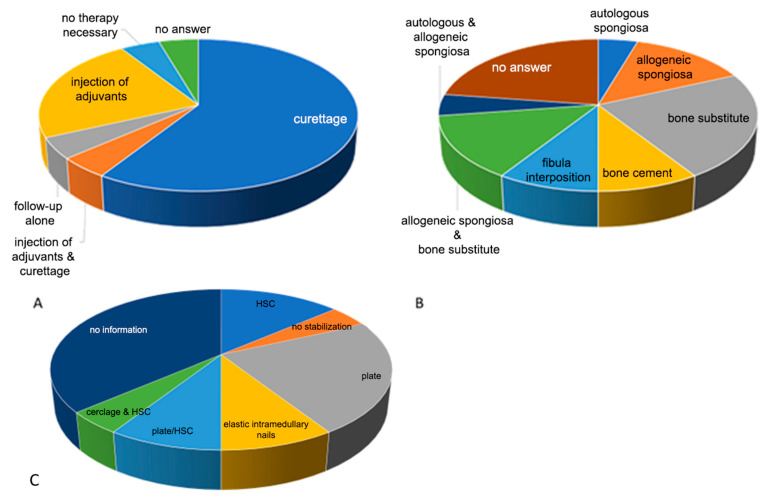
(**A**) As local therapy, the following were recommended by the respondents: curettage, 68.2%; adjuvant injection and curettage, 4.5%; follow-up alone, 4.5%; injection of adjuvants, 22.7%; no therapy necessary, 4.5%; and no information, 4.5%. (**B**) For defect filling, autologous spongiosa was recommended by 4.5% of the respondents; allogeneic spongiosa, by 13.6%; bone substitute, by 27.3%; bone cement, by 9%; fibula interposition, by 9%; allogeneic spongiosa and bone substitute, by 13.6%; and autologous and allogeneic spongiosa, by 4.5%. Of the respondents, 22.7% gave no answer. (**C**) For osteosynthesis/stabilisation, most of the 22 hospitals recommended surgical stabilisation (40.9%), 9% of the clinics preferred immobilisation with an HSC only; 9% did not commit themselves and recommended a procedure (plate osteosynthesis or HSC as a secondary alternative) depending on the findings and completion of the cyst removal and filling; and 4.5% saw no need for immobilisation or stabilisation. Surgical stabilisation using plate osteosynthesis (27%), elastic intramedullary nails (9%), or cerclage and additional HSC immobilisation were mentioned. Of the participating hospitals, 36.4% did not provide any information on this question.

## Data Availability

The data that support the findings of this study are available from Roman Michalik and Heide Delbrück, but restrictions apply to the availability of these data, which were used under license for the present study and so are not publicly available.

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
