# Peer review of "Stabilisation of Pathologic Proximal Femoral Fracture near the Growth Plate with Use of a Locking Plate and Transphyseal Screws"

_children, 2022, doi:10.3390/children9121932_

Round 1

Reviewer 1 Report

Very interesting concept that growth cartilage could be fixed with screw without any concern for the growth of the bone.

Reviewer 2 Report

The authors presented an article named: "Treatment of aneurysmal bone cysts in children near the growth plate of femoral neck and proximal femur. A national survey, presentation of a treatment algorithm including a stabilization technique using locking plate and transphyseal screws."

This is a national survey with a proposed algorithm and a case report of a 20month child. 

This is a nice article, well written with minor spelling mistakes, doing a bit of controversial subject.

The conclusion should not contain repetition and explaining the AIMS.

My final impression and remark is that the conclusion should be modified and changed. This cannot be a widely accepted method for all (as I too like, and advocate the one-in-three therapy options) meaning for younger children because the shown case report of a 20month old child is not a good example. Transphyseal screws may or may not significantly damage the growth plate and this remains to be seen. It is not the same to do this in a 14-year-old child and a 2-year-old.

Additionally, it is needed far larger follow-up for this child. At least 5 years to see the possible complications and growth differences.

This is a nice case report with a done survey but this cannot be used for serious conclusions without a larger patient sample and a far more larger follow-up period.

Should be stated as a limitations also.
